# Cost-effectiveness of internet-based vestibular rehabilitation with and without physiotherapy support for adults aged 50 and older with a chronic vestibular syndrome in general practice

Vincent A van Vugt  ,[1,2] Judith E Bosmans,[2,3] Aureliano P Finch,[2,3] Johannes C van der Wouden,[1,2] Henriëtte E van der Horst,[1,2] Otto R Maarsingh[1,2]

► Prepublication history and additional materials for this paper is available online. To view these files, please visit the journal online (http://dx.doi.org/10.1136/bmjopen-2019-035583).

¹Department of General Practice, Amsterdam UMC - Location VUMC, Amsterdam, The Netherlands
²Amsterdam Public Health Research Institute, Amsterdam, The Netherlands
³Department of Health Sciences, Vrije Universiteit Amsterdam, Amsterdam, The Netherlands

**Correspondence to**
Vincent A van Vugt;
v.vanvugt@amsterdamumc.nl

## ABSTRACT

**Objectives** To evaluate the cost-effectiveness of stand-alone and blended internet-based vestibular rehabilitation (VR) in comparison with usual care (UC) for chronic vestibular syndromes in general practice.

**Design** Economic evaluation alongside a three-armed, individually randomised controlled trial.

**Setting** 59 Dutch general practices.

**Participants** 322 adults, aged 50 years and older with a chronic vestibular syndrome.

**Interventions** Stand-alone VR consisted of a 6-week, internet-based intervention with weekly online sessions and daily exercises. In blended VR, this intervention was supplemented with face-to-face physiotherapy support. UC group participants received usual general practice care without restrictions.

**Main outcome measures** Societal costs, quality-adjusted life years (QALYs), Vertigo Symptom Scale—Short Form (VSS-SF), clinically relevant response (≥3 points VSS-SF improvement).

**Results** Mean societal costs in both the stand-alone and blended VR groups were statistically non-significantly higher than in the UC group (mean difference (MD) €504, 95% CI −1082 to 2268; and €916, 95% CI −663 to 2596). Both stand-alone and blended VR groups reported non-significantly more QALYs than the UC group (MD 0.02, 95% CI −0.00 to 0.04; and 0.01, 95% CI −0.01 to 0.03), and significantly better VSS-SF Scores (MD 3.8 points, 95% CI 1.7 to 6.0; and 3.3 points, 95% CI 1.3 to 5.2). For stand-alone VR compared with UC, the probability of cost-effectiveness was 0.95 at a willingness-to-pay ratio of €24 161/QALY, €600/point improvement in VSS-SF and €8000/clinically relevant responder in VSS-SF. For blended VR versus UC, the probability of cost-effectiveness was 0.95 at a willingness-to-pay ratio of €123 335/QALY, €900/point improvement in VSS-SF and €24 000/clinically relevant responder in VSS-SF.

**Conclusion** Stand-alone and blended internet-based VR non-significantly increased QALYs and significantly reduced vestibular symptoms compared with UC, while costs in both groups were non-significantly higher. Stand-alone VR has the highest probability to be cost-effective compared with UC.

### Strengths and limitations of this study

► This is the first study to evaluate the cost-effectiveness of stand-alone internet-based vestibular rehabilitation and blended vestibular rehabilitation versus usual care for patients with a chronic vestibular syndrome in general practice.

► The economic evaluation was conducted based on the data of 322 participants aged 50 years and older with a chronic vestibular syndrome.

► Strengths of our study are the societal perspective on costs, the use of multiple outcome measures and the use of a sensitivity analysis to test robustness of our findings.

► Limitations of the study are that it was powered for the primary outcome measure Vertigo Symptom Scale—Short Form and not for costs or quality-adjusted life years, and that the follow-up was limited to 6 months.

**Trial registration number** The Netherlands Trial Register NTR5712.

## INTRODUCTION

General practitioners (GPs) frequently encounter patients with vestibular symptoms (ie, vertigo, dizziness, vestibulovisual and postural symptoms).[1–3] The 1-year prevalence of vestibular symptoms is approximately 20% in population-based studies.[1] Patients with vestibular symptoms are often unable to work, make frequent use of healthcare services and have an increased risk of falling.[1 4 5] The financial burden of these symptoms is therefore substantial. In a recent German study, the yearly healthcare costs were over €800 higher in patients with vestibular symptoms than in patients without vestibular symptoms.[5] Through assessment of timing and triggers of symptoms, patients can be classified as having

either an acute vestibular syndrome, episodic vestibular syndrome or chronic vestibular syndrome.[6–8] In chronic vestibular syndromes, patients experience vestibular symptoms with features suggestive of persistent vestibular system dysfunction for months to years. The preferred treatment for chronic vestibular syndromes, according to clinical guidelines from the USA,[9 10] the UK[11] and The Netherlands,[12] is vestibular rehabilitation (VR). VR is an exercise-based treatment developed to reduce vestibular symptoms by gradually stimulating the vestibular system.[13 14] There is moderate to strong evidence that VR can safely and effectively reduce vestibular symptoms in both unilateral and bilateral peripheral vestibular dysfunction.[14 15]

Despite the scientific evidence and recommendations in guidelines, VR is still underused in general practice. Surveys among Dutch[16] and English[17] GPs indicate that less than 10% uses VR. Developing new ways to deliver VR in general practice may help to implement this effective treatment in daily practice. Internet interventions have several advantages over other forms of treatment: they are inexpensive, easily accessible and can be easily personalised according to the needs of individual patients. Recently, the University of Southampton transformed the content of a VR booklet[18] that was shown to be (cost-)effective compared with usual care, into an internet-based VR intervention.[19] In a randomised controlled trial in the UK,[20] this stand-alone internet-based VR intervention effectively reduced vestibular symptoms compared with usual care. Combining an internet-based intervention with face-to-face support by a healthcare professional is called blended care.[21] Stand-alone internet-based interventions are generally less expensive than blended internet-based interventions, but the risk of non-adherence is also higher.[21–23] We therefore developed a blended internet-based VR intervention by adding physiotherapy support to the British internet-based VR intervention. We conducted a pragmatic, three-armed, randomised controlled trial in Dutch general practice to investigate the clinical effectiveness and cost-effectiveness of stand-alone and blended internet-based VR compared with usual care.[24] As we recently reported, both stand-alone and blended internet-based VR led to a clinically relevant and statistically significant decrease in vestibular symptoms compared with usual care.[25] The objective of the present substudy is to evaluate the cost-effectiveness of stand-alone and blended internet-based VR versus usual care for general practice patients aged 50 and older with a chronic vestibular syndrome.

## METHODS
### Design
This economic evaluation was conducted alongside a pragmatic, three-armed, randomised controlled trial in general practice. In the trial, we compared stand-alone and blended internet-based VR with usual care in patients aged 50 and older with a chronic vestibular syndrome.

A detailed description of the study protocol[24] and the results of the clinical effectiveness analysis[25] can be found in previous publications. For the reporting of the cost-effectiveness analysis, we follow the recommendations in the Consolidated Health Economic Evaluation Reporting Standards statement.[26]

### Participants
In short, we recruited participants from 59 general practices in The Netherlands. Patients with chronic vestibular syndrome according to the International Classification of Vestibular Disorders were eligible for the study.[6 7] Chronic vestibular syndrome was defined as vestibular symptoms at time of inclusion that had been present for at least 1 month and that were exacerbated or triggered by performing head movement. Further inclusion criteria were: age 50 years and older; good command of the Dutch language; access to the internet and an email account. Participants with an identified non-vestibular cause of dizziness, medical contraindications for making the required head movements (eg, severe cervical arthrosis), serious comorbid conditions precluding participation in an exercise programme or current enrolment in another study were excluded.

### Interventions
#### Stand-alone internet-based VR (stand-alone VR)
Vertigo Training, the internet-based VR intervention we used in this trial, is a Dutch translation of the internet-based VR intervention developed by the University of Southampton. The intervention lasted 6 weeks and consisted of 6 weekly online sessions with daily VR exercises. In the first session, video demonstrations and written instructions are used to teach participants the six core VR exercises. During the intervention period, participants were asked to perform these exercises for 10 min two times per day. Every week, the participant logged into Vertigo Training to self-report the level of vestibular symptoms caused by each of the six exercises. Vertigo Training used this information to produce a VR exercise prescription for the coming week, tailored to the individual needs and capabilities of the participant. In addition, Vertigo Training also provides information and advice on coping and symptom control strategies which are described in more detail elsewhere.[25]

#### Blended internet-based VR with physiotherapist support (blended VR)
Supplemental to the Vertigo Training intervention, participants in the blended VR group were visited twice at home by a trained physiotherapist. These supportive physiotherapy sessions occurred in weeks one and three of the 6-week intervention period and lasted for 45 min each.

#### Usual care
Participants in the usual care group received the standard level of care provided by their own GP without restrictions.

## Measures

We assessed quality of life with the most widely used preference-based quality-of-life instrument in economic evaluations,[27] the 5-level EuroQol questionnaire (EQ-5D-5L), at baseline, and 3 and 6 months of follow-up.[28] EQ-5D-5L health states were converted to utility scores using the Dutch EQ-5D-5L tariff.[29] For all five dimensions, a subtraction is done from the utility score if the participant experiences any problems on that domain. We calculated quality-adjusted life years (QALYs) by multiplying the utility of a specific health state with the time spent in that health state. Transitions between health states were linearly interpolated. One QALY is equivalent to one life-year in perfect health. A gain in QALYs can therefore be interpreted as an improvement in the quantity and/or quality of life.

We measured vestibular symptoms by the Vertigo Symptom Scale—Short Form (VSS-SF)[30 31] at baseline, and 3 and 6 months of follow-up. The VSS-SF measures the frequency of 15 vestibular symptoms on a scale from 0 (no symptoms) to 4 (symptoms most days) during the past month (total range 0–60 points). Improvement can reflect either fewer or less frequent symptoms. In accordance with previous studies,[18 20 32 33] we defined a decrease of three points or more on the VSS-SF between baseline and 6 month measurement as a clinically relevant response. In this cost-effectiveness study, we used the VSS-SF Scores at 6 months follow-up and the number of participants with a clinically relevant improvement during the 6-month trial period as measures of effectiveness.

We measured costs from a societal perspective, using the iMTA Medical and Productivity Cost Questionnaires at 3 and 6 months of follow-up.[34 35] Both questionnaires had a recall period of 3 months. The societal costs included healthcare costs (primary care, secondary care, medication and home care), informal care costs (ie, costs related to help from family and friends) and lost productivity costs. Lost productivity costs consisted of costs related to absenteeism from paid and unpaid work (eg, household activities or voluntary work), and presenteeism costs. Presenteeism is defined as reduced efficiency due to health problems while at work.[36]

We valued healthcare utilisation by using Dutch standard costs if available.[37] Otherwise, we used tariffs of professional organisations or healthcare providers themselves. We used data from the Dutch Healthcare Institute to value medication use.[38] We estimated costs of absenteeism from paid work by using the friction cost approach[39] that assumes that sick employees are replaced after a certain period of time (the friction period, ie, 12 weeks) after which there are no productivity losses anymore. We calculated productivity losses associated with paid work with gender-specific average wage rates of the Dutch population.[37] We assessed presenteeism by measuring participants' efficiency while at work on a numerical rating scale ranging from 0 ('I could not do anything') to 10 ('I did as much as always'), that is, the efficiency score. To calculate lost productivity hours due to presenteeism, we multiplied one minus the efficiency score with the number of days of reduced efficiency, multiplied by the number of working hours per day. We then used gender-specific average wage rates to convert the lost productivity hours due to presenteeism to lost productivity costs. To calculate lost productivity costs associated with unpaid work and informal care costs, we used a shadow price for a legally employed cleaner.[37]

## Cost-effectiveness analyses

We performed the economic evaluation from a societal perspective. We compared QALYs and VSS-SF Scores at 6 months pairwise between the two active treatment groups (stand-alone and blended VR) and usual care. Discounting was not necessary, because follow-up was shorter than 1 year.

We performed the analyses according to the intention to treat principle. We imputed missing cost and effect data with multiple imputation by chained equations with predictive mean matching to account for the skewed distribution of costs.[40] The advantage of multiple imputation over single imputation methods is that the uncertainty around the imputed values is also taken into account. The predictive mean matching ensured that only observed values can be imputed. The number of imputed datasets was increased until the fraction of missing information was smaller than 5%.[41] The imputation model included variables that differed at baseline, differed between participants with and without complete follow-up, or that were associated with the outcomes (clinical effects and costs). In addition, we included all variables of the analysis models in the imputation model. We performed analyses separately for each imputed dataset and then pooled applying Rubin's rules.[42]

We estimated differences in costs and effects with bivariate regression in which two separate regression models are specified, while correlation between costs and effects is maintained through correlated error terms. This allows for specification of separate covariates for costs and effects in the regression models. We used bias-corrected accelerated bootstrapping with 5000 replications to estimate the uncertainty surrounding differences in costs and effects. We calculated incremental cost-effectiveness ratios (ICERs) by dividing the difference in costs by the difference in effects. ICERs indicate the investment that is needed to gain one unit of effect extra in the intervention group compared with control. Uncertainty surrounding the ICER is shown in cost-effectiveness planes.[43] The probability of cost-effectiveness was calculated by determining the proportion of bootstrapped cost–effect pairs below the willingness-to-pay threshold for each possible willingness-to-pay threshold. By estimating cost-effectiveness acceptability (CEA), we subsequently combined statistical uncertainty with decision uncertainty curves to show the probability that the intervention is cost-effective compared with usual care for different willingness-to-pay ratios.[44] The willingness-to-pay ratio is defined as the maximum amount of money that society

is willing to pay to gain one additional unit of effect. We employed the commonly used threshold of €20,000/QALY gained for appraisal of new health technologies.[45]

We conducted three sensitivity analyses to assess the robustness of the findings. In the first sensitivity analysis (SA1), we conducted the economic evaluation from the perspective of the healthcare system, meaning that only healthcare costs are included, as is recommended in some other countries, such as the UK, Belgium and Germany. In the second sensitivity analysis (SA2), we coded costs of extreme outliers (five participants who were admitted to the intensive care unit during the trial for reasons unrelated to the trial) as missing so they would be imputed in the multiple imputation procedure. In the third sensitivity analysis (SA3), we excluded secondary care and medication costs. We chose to conduct SA3 because hospitalisations and expensive medications could have had a large influence on the main analysis due to relatively high costs, even though they were all judged to be unrelated to vestibular symptoms.

## Patient and public involvement

Patients played an important role in the development of Vertigo Training. Detailed feedback by patients with vestibular symptoms on the content, usability and Dutch translation in prototype versions led to some amendments of the online intervention. No patients advised on interpretation of the results, nor were they involved in writing the manuscript. A lay summary of the research findings will be distributed to all participants in the study and the results will be disseminated to the relevant patient community.

## RESULTS

### Participants

We provide an overview of patient enrolment, allocation and follow-up in online supplemental figure 1. Participants were recruited between June 2017 and July 2018. We randomised 322 participants at baseline: 98 participants were allocated to the stand-alone VR group, 104 to the blended VR group and 120 to the usual care group. The baseline characteristics of participants are shown in online supplemental table 1.

### Effects

Table 1 presents the pooled mean effects and costs in the three intervention groups. The mean QALY was 0.43 for stand-alone VR participants, 0.41 for blended VR participants and 0.41 for usual care participants. The differences of stand-alone and blended VR compared with usual care in QALYs were not statistically significant (mean difference 0.02, 95% CI −0.00 to 0.04; and 0.01, 95% CI −0.01 to 0.03). Both

**Table 1** Pooled mean effect and cost outcomes (SE) stratified for treatment group and differences in mean effect and cost outcomes (95% CI) for the intervention groups compared with usual care

| Outcome | Stand-alone VR n=98 | Blended VR n=104 | Usual care n=120 | Difference stand-alone VR versus usual care | Difference blended VR versus usual care |
|---|---|---|---|---|---|
| **Effects** | | | | | |
| QALY | 0.43 (0.008) | 0.41 (0.008) | 0.41 (0.008) | 0.02 (−0.001 to 0.04) | 0.01 (−0.01 to 0.03) |
| VSS-SF | 8.1 (0.91) | 8.5 (0.70) | 11.4 (0.95) | 3.8 (1.7 to 6.0)* | 3.3 (1.3 to 5.2)* |
| Response | 0.72 (0.05) | 0.61 (0.05) | 0.45 (0.95) | 0.27 (0.13 to 0.40) | 0.16 (0.02 to 0.30) |
| **Costs** | | | | | |
| Intervention | 39 (0) | 155 (5) | 0 (0) | 39 | 155 |
| Total healthcare | 1379 (377) | 1116 (229) | 901 (162) | 478 (−100 to 1514) | 215 (−272 to 836) |
| Primary care | 304 (42) | 362 (59) | 315 (37) | −11 (−112 to 97) | 47 (−72 to 200) |
| Complementary medicine | 22 (7) | 8 (3) | 29 (11) | −7 (−37 to 12) | −21 (−54 to −6) |
| Outpatient care | 104 (19) | 147 (78) | 116 (20) | −12 (−62 to 37) | 31 (−62 to 344) |
| Admissions | 682 (350) | 22 (19) | 129 (95) | 553 (102 to 1563) | −107 (−471 to 2) |
| Medication | 209 (78) | 352 (172) | 171 (57) | 38 (−105 to 288) | 181 (−46 to 817) |
| Home care | 58 (22) | 224 (107) | 140 (43) | −82 (−193 to −5) | 84 (−59 to 460) |
| Informal care | 88 (30) | 187 (93) | 112 (38) | −24 (−123 to 56) | 75 (−53 to 392) |
| Total lost productivity | 2061 (538) | 2521 (541) | 2049 (443) | 12 (−1260 to 1439) | 472 (−846 to 1825) |
| Absenteeism | 241 (133) | 931 (331) | 486 (199) | −245 (−776 to 190) | 445 (−186 to 1337) |
| Presenteeism | 10 (3) | 14 (5) | 15 (5) | −5 (−19 to 4) | −1 (−15 to 12) |
| Unpaid work | 1810 (518) | 1576 (229) | 1548 (361) | 262 (−807 to 1618) | 28 (−981 to 1069) |
| Total societal | 3567 (701) | 3979 (667) | 3063 (520) | 504 (−1082 to 2268) | 916 (−663 to 2596) |

*Due to a different method of analysis VSS-SF Scores slightly differ from the previously reported clinical effectiveness analysis.[25]

QALY, quality-adjusted life years; Response, percentage of participants with a decrease of ≥3 points in VSS-SF between baseline and 6 months; VR, vestibular rehabilitation; VSS-SF, Vertigo Symptom Scale—Short Form, range 0–60, clinically relevant difference 3 points.

van Vugt VA, et al. BMJ Open 2020;**10**:e035583. doi:10.1136/bmjopen-2019-035583

stand-alone and blended VR participants reported statistically significantly lower VSS-SF Scores at 6 months follow-up compared with usual care participants (mean difference 3.8 points, 95% CI 1.7 to 6.0; and 3.3 points, 95% CI 1.3 to 5.2). A clinically relevant VSS-SF response was seen significantly more often in both stand-alone and blended VR groups than in the usual care group (difference in percentage of participants with clinically relevant response 27%, 95% CI 13% to 40%; and 16%, 95% CI 2% to 30%). Overall, differences in effects for stand-alone VR versus usual care were larger than for blended VR versus usual care.

## Costs

Total societal costs in the stand-alone group were €3567 (SE 701), in the blended VR group €3979 (SE 667) and in the usual care group €3063 (SE 520). However, uncertainty surrounding these differences was substantial. The cost difference between stand-alone VR and usual care was smaller (ie, mean difference €504, 95% CI –1082 to 2268) than between blended VR and usual care (mean difference €916, 95% CI –663 to 2596). Healthcare costs in both the stand-alone and blended VR groups were statistically non-significantly higher than in the usual care group (mean difference €478, 95% CI –100 to 1514; and €215, 95% CI –272 to 836). Total lost productivity costs in the stand-alone VR

and usual care groups were similar (mean difference €12, 95% CI –1260 to 1439). Total lost productivity costs in the blended VR group were considerably—but not significantly—higher than in the usual care group (mean difference €472, 95% CI –846 to 1825).

## Cost-effectiveness main analysis

Table 2 presents the results of the cost-effectiveness analyses for stand-alone VR compared with usual care. For QALYs, the ICER was €24 161/QALY gained for stand-alone VR in comparison with usual care. The probability of cost-effectiveness was 0.28 at a willingness-to-pay ratio of €0/QALY gained. At a willingness-to-pay ratio of €20 000/QALY gained the probability of cost-effectiveness was 0.47 for stand-alone VR versus usual care. For VSS-SF, the ICER was 132, indicating that €132 needs to be invested to gain one point of improvement in VSS-SF in the stand-alone VR group compared with usual care. The probability that stand-alone VR is cost-effective compared with usual care was 0.28 and 0.95 at willingness-to-pay ratios of 0 and €600/point improvement in VSS-SF, respectively (figure 1). The ICER for response indicates that one additional participant with a clinically relevant response on the VSS-SF (≥3 points improvement) requires an investment of on average €1895 in the stand-alone VR group compared with the

**Table 2** Cost-effectiveness outcomes for stand-alone VR compared with usual care

| Outcome | ΔC (95% CI) | ΔE (95% CI) | ICER | CE plane (%) | | | |
|---|---|---|---|---|---|---|---|
| | | | | NE | SE | SW | NW |
| Main analysis—societal perspective | | | | | | | |
| QALY | 504 (–1064 to 2303) | 0.02 (–0.001 to 0.04) | 24 161 | 69 | 28 | 0 | 3 |
| VSS-SF | 504 (–1052 to 2294) | 3.8 (1.7 to 6.0) | 132 | 72 | 28 | 0 | 0 |
| Response* | 504 (–1057 to 2282) | 0.27 (0.13 to 0.40) | 1895 | 72 | 28 | 0 | 0 |
| SA1—healthcare perspective | | | | | | | |
| QALY | 478 (–103 to 1510) | 0.02 (–0.001 to 0.04) | 22 936 | 86 | 11 | 0 | 3 |
| VSS-SF | 478 (–94 to 1527) | 3.8 (1.7 to 6.0) | 126 | 89 | 11 | 0 | 0 |
| Response* | 478 (–104 to 1520) | 0.27 (0.13 to 0.40) | 1799 | 89 | 11 | 0 | 0 |
| SA2—societal perspective, outliers recoded as missings | | | | | | | |
| QALY | 241 (–1194 to 1838) | 0.02 (–0.001 to 0.04) | 11 388 | 58 | 39 | 0 | 3 |
| VSS-SF | 241 (–1184 to 1851) | 3.7 (1.6 to 5.7) | 66 | 60 | 40 | 0 | 0 |
| Response* | 241 (–1195 to 1832) | 0.27 (0.13 to 0.40) | 910 | 61 | 39 | 0 | 0 |
| SA3—societal perspective, only costs related to vestibular symptoms | | | | | | | |
| QALY | –75 (–1425 to 1450) | 0.02 (–0.001 to 0.04) | –3603 | 43 | 54 | 1 | 2 |
| VSS-SF | –75 (–1430 to 1463) | 3.8 (1.7 to 6.0) | –20 | 45 | 55 | 0 | 0 |
| Response* | –75 (–1432 to 1453) | 0.27 (0.13 to 0.40) | –283 | 45 | 55 | 0 | 0 |

*Response was defined as ≥3 points improvement on the Vertigo Symptom Scale—Short Form after 6 months.
ΔC, cost difference between stand-alone VR and usual care; CE, cost-effectiveness; ΔE, effect difference between stand-alone VR and usual care; ICER, incremental cost-effectiveness ratio; NE, northeast (more expensive and more effective); NW, northwest (more expensive and less effective); QALY, quality-adjusted life years; SA1, sensitivity analysis with only healthcare costs included; SA2, sensitivity analysis with extreme outliers recoded as missings; SA3, sensitivity analysis with secondary care and medication costs excluded; SE, southeast (less expensive and more effective); SW, southwest (less expensive and less effective); VR, vestibular rehabilitation; VSS-SF, Vertigo Symptom Scale—Short Form.

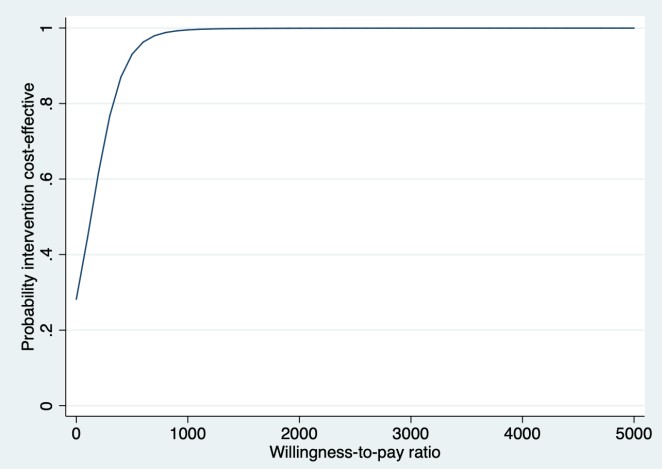

**Figure 1** Main analysis—societal perspective. Cost-effectiveness acceptability curve for the Vertigo Symptom Scale—Short Form comparing stand-alone VR with usual care. VR, vestibular rehabilitation.

usual care group. At ceiling ratios of 0 and €8000/additional responder, the probability that stand-alone care is cost-effective in comparison with usual care was 0.28 and 0.95, respectively (see online supplemental figure 2).

In table 3 the results of the cost-effectiveness analyses for blended VR in comparison with usual care are shown. For QALYs, the ICER was 123 335 (the large ICER is caused by the small difference in effects), meaning that €123 335 needs to be invested to gain 1 QALY in the blended VR group compared with the usual care group. The probability of cost-effectiveness was 0.14 at a ceiling ratio of 0 €/QALY gained and increased to 0.21 at a ceiling ratio of €20 000/QALY gained. For VSS-SF, the ICER was 280 for blended VR compared with usual care, indicating that to gain one point of improvement in VSS-SF, €280 needs to be invested in blended VR compared with usual care. The CEA curve (figure 2) shows that the probability of cost-effectiveness rapidly increases from 0.14 at a willingness-to-pay ratio of €0/point improvement in VSS-SF to 0.95 at a willingness-to-pay ratio of €900/point of improvement in VSS-SF. The ICER for response was 5599, indicating that one patient extra with a clinically relevant response on the VSS-SF (≥3 points improvement) is associated with an investment of €5599 for blended VR in comparison with usual care. The CEA curve (see online supplemental figure 3) indicates that the probability of cost-effectiveness is 0.14 and 0.95 at ceiling ratios of 0 and €24 000/responder extra, respectively.

### Sensitivity analyses

Table 2 presents the results of the sensitivity analyses for stand-alone VR compared with usual care and table 3

| Outcome | ΔC (95% CI) | ΔE (95% CI) | ICER | NE | SE | SW | NW |
|---|---|---|---|---|---|---|---|
| **Table 3** Cost-effectiveness outcomes for blended VR compared with usual care | | | | | | | |
| | | | | **CE plane (%)** | | | |
| | | | | NE | SE | SW | NW |
| *Main analysis—societal perspective* | | | | | | | |
| QALY | 916 (–660 to 2579) | 0.007 (–0.01 to 0.03) | 123 335 | 62 | 12 | 1 | 25 |
| VSS-SF | 916 (–655 to 2610) | 3.3 (1.3 to 5.2) | 280 | 86 | 14 | 0 | 0 |
| Response* | 916 (–658 to 2583) | 0.16 (0.02 to 0.30) | 5599 | 85 | 13 | 0 | 0 |
| *SA1—healthcare perspective* | | | | | | | |
| QALY | 215 (–273 to 820) | 0.007 (–0.01 to 0.03) | 28 848 | 55 | 19 | 3 | 23 |
| VSS-SF | 215 (–263 to 828) | 3.3 (1.4 to 5.2) | 65 | 78 | 22 | 0 | 0 |
| Response* | 215 (–263 to 832) | 0.16 (0.02 to 0.30) | 1310 | 77 | 22 | 0 | 1 |
| *SA2—societal perspective, outliers recoded as missings* | | | | | | | |
| QALY | 1140 (–404 to 2811) | 0.007 (–0.01 to 0.03) | 156 954 | 66 | 8 | 1 | 25 |
| VSS-SF | 1140 (–404 to 2811) | 3.1 (1.3 to 5.0) | 366 | 91 | 9 | 0 | 0 |
| Response* | 1140 (–401 to 2789) | 0.16 (0.02 to 0.30) | 6988 | 91 | 8 | 0 | 1 |
| *SA3—societal perspective, only costs related to vestibular symptoms* | | | | | | | |
| QALY | 810 (–653 to 2400) | 0.007 (–0.01 to 0.03) | 109 121 | 61 | 13 | 1 | 24 |
| VSS-SF | 810 (–653 to 2406) | 3.3 (1.4 to 5.2) | 248 | 85 | 15 | 0 | 0 |
| Response* | 810 (–650 to 2410) | 0.16 (0.02 to 0.30) | 4954 | 84 | 15 | 0 | 1 |

*Response was defined as ≥3 points improvement on the Vertigo Symptom Scale – Short Form after 6 months.
ΔC, cost difference between stand-alone VR and usual care; CE, cost-effectiveness; ΔE, effect difference between stand-alone VR and usual care; ICER, incremental cost-effectiveness ratio; NE, northeast (more expensive and more effective); NW, northwest (more expensive and less effective); QALY, quality-adjusted life-years; SA1, sensitivity analysis with only healthcare costs included; SA2, sensitivity analysis with extreme outliers recoded as missings; SA3, sensitivity analysis with secondary care and medication costs excluded; SE, southeast (less expensive and more effective); SW, southwest (less expensive and less effective); VR, vestibular rehabilitation; VSS-SF, Vertigo Symptom Scale—Short Form.

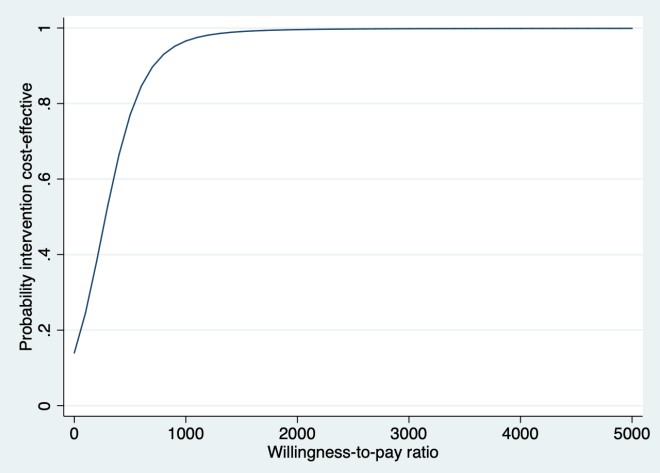

**Figure 2** Main analysis—societal perspective. Cost-effectiveness acceptability curve for the Vertigo Symptom Scale—Short Form comparing blended VR with usual care. VR, vestibular rehabilitation.

shows the results of the sensitivity analyses for blended VR compared with usual care. In SA1, in which the analysis was performed from the healthcare perspective, healthcare costs in the stand-alone VR group were higher than in the usual care group (mean difference €478, 95% CI –100 to 1514), although the differences in healthcare costs were smaller than the differences in societal costs (mean difference €504, 95% CI –1064 to 2303). There was less statistical uncertainty surrounding healthcare costs compared with societal costs. In this sensitivity analysis, the ICERs for stand-alone VR compared with usual care with regard to QALYs (22936), VSS-SF (126) and response (1799) were all slightly lower than in the main analysis (QALYs 24161; VSS-SF 132 and response 1895). Healthcare costs in the blended VR group were higher than in the usual care group, but smaller than the difference in societal costs in the main analysis (mean difference €215, 95% CI –272 to 836; versus €916, 95% CI –663 to 2596). As a result, in the comparison between blended VR and usual care, the ICERs for QALYs (28848), VSS-SF (65) and response (1310) were substantially lower than in the main analysis (QALYs 123335; VSS-SF 280 and response 5599).

In SA2, where we coded the extreme outliers as missing and imputed their data, the societal cost difference between stand-alone VR and usual care decreased from €504 (95% CI –1082 to 2268) to €454 (95% CI –1019 to 2399), as shown in online supplemental file 2. For stand-alone VR in comparison to usual care, the ICERs for QALYs, VSS-SF and response were all markedly lower and the probability of cost-effectiveness at a willingness-to-pay ratio of €0€/unit of effect extra increased from 0.28 in the main analysis to 0.39 in SA2 (table 2). For blended VR compared with usual care, the societal cost difference became larger and the probabilities of cost-effectiveness lower (table 3).

In SA3, where we excluded costs not related to vestibular symptoms, results become more positive due to smaller differences in costs between the stand-alone and

blended VR groups and usual care. For blended VR in comparison with usual care, this effect is relatively small (cost difference €810, 95% CI –653 to 2400; instead of €916, 95% CI –663 to 2596), resulting in similar probabilities of cost-effectiveness as in the main analysis (table 3). However, for stand-alone VR in comparison with usual care the cost difference inverts (table 2; -€75, 95% CI –1425 to 1450; instead of €504, 95% CI –1064 to 2303). As a result, in this analysis stand-alone VR is dominant in comparison with usual care (ie, more effective and less expensive). CEA curves indicate that the probability of cost-effectiveness is 0.72 at a ceiling ratio of €20000/QALY (see online supplemental figure 4), 0.95 at ceiling ratios of €350/point of improvement in VSS-SF (see online supplemental figure 5) and 0.95 at ceiling ratios of €4600/additional responder based on the VSS-SF (see online supplemental figure 6).

## DISCUSSION
### Principal findings
Stand-alone and blended VR resulted in a statistically non-significant gain in QALYs and significantly reduced vestibular symptoms compared with usual care. Total societal costs of both stand-alone and blended VR were statistically non-significantly higher than the costs of usual care. Since stand-alone VR was more effective and less expensive than blended VR, the probability for stand-alone VR to be cost-effective in comparison with usual care was higher. In a sensitivity analysis, where we excluded costs that were unlikely to be related to vestibular symptoms or the use of internet-based VR, stand-alone VR became dominant (ie, more effective and less expensive) over usual care.

### Comparison with existing literature
This is the first study to investigate the cost-effectiveness of internet-based VR for patients with a chronic vestibular syndrome in general practice. This internet-based VR intervention is based on the content of a VR booklet that was previously shown to be highly cost-effective compared with usual care in a randomised controlled trial.[18] Even though both internet-based and booklet-based VR interventions were inexpensive, the total costs of participants in our stand-alone VR and blended VR groups were much higher than in the study evaluating the VR booklet. This difference in costs between the trials probably occurred due to a more narrow definition of costs in the booklet VR trial. In the booklet VR trial, costs only comprised intervention costs and resource use of vestibular-related healthcare. A blinded researcher examined the medical records of participants to assess whether contacts were related to vestibular symptoms. In our trial, we analysed societal costs, which included all healthcare costs (regardless of the reason of contact), informal care costs and lost productivity costs. Total societal costs in the internet-based VR groups were considerably higher than in the usual care group. The largest contributor to the difference in costs

between stand-alone VR and usual care was hospital admission costs. During the trial, 7 stand-alone VR participants, 2 blended VR participants and 5 usual care participants were admitted to the hospital. Each hospitalisation was evaluated by contacting the participant and/or GP, and none of the hospitalisations were judged to be related to vestibular symptoms and/or internet-based VR. Because care in primary care is generally less expensive, this skewness in costly hospitalisations could have strongly affected the results of the cost-effectiveness analysis. Therefore, we excluded secondary care and medication costs that were likely to be unrelated to vestibular symptoms and/or internet-based VR in the third sensitivity analysis. In this analysis, stand-alone VR was dominant over usual care (ie, more effective and less costly). For blended VR, the largest difference in costs with usual care was not seen in admissions but in loss of productivity. These relatively high absenteeism rates cannot easily be explained, but contact with a physiotherapist might have changed the way participants prioritised daily activities while experiencing vestibular symptoms. Blended VR participants may have deliberately reduced paid and unpaid work to decrease emotional distress, because they learnt this can exacerbate their vestibular symptoms.[46]

### Strengths and limitations

Strengths of our study are the pragmatic design of our randomised controlled trial, the societal perspective on costs, the use of multiple outcome measures (QALYs, VSS-SF, clinically relevant response) and conduct of three sensitivity analyses in addition to the main analysis to assess the robustness of the results.

There are also several limitations. First, the trial was powered for the primary outcome measure VSS-SF and not for costs or QALYs. This is reflected in a substantial uncertainty surrounding the relatively large cost differences between groups that did not reach statistical significance. Second, the effectiveness and costs were measured over 6 months. Based on this study, we cannot ascertain the long-term cost-effectiveness of stand-alone and blended VR versus usual care. Nevertheless, an increase in cost-effectiveness might occur in the long term. Previous VR trials have also shown effectiveness at twelve months,[14 18] and costs for internet-based VR are not expected to increase during a longer follow-up period. Third, the differences measured in QALYs between the intervention groups and usual care were not significant, unlike the differences in VSS-SF Scores. The EQ-5D-5L, a generic utility measure, may not have been sensitive enough to capture changes in quality of life in our population. Certain domains that are important for patients with chronic vestibular syndromes, such as the disease's unpredictability and its impact on social and role functioning, are not measured by the EQ-5D-5L.[47 48] We also used the Dizziness Handicap Inventory[49 50] in our trial, an outcome measure that specifically quantifies the impact on daily life by vestibular symptoms. We did find significant differences in Dizziness Handicap Inventory

Scores favouring stand-alone VR and blended VR over usual care (adjusted mean difference at 6 months: −4.9 points, 95% CI −8.4 to −1.3; and −4.5 points, 95% CI −8.0 to −0.9).[25] Using the EQ-5D-5L might have caused us to underestimate the impact of stand-alone and blended VR on quality of life.

### Conclusions and implications for research and/or practice

Stand-alone and blended internet-based VR both led to a statistically non-significant gain in QALYs and a significant reduction of vestibular symptoms, but the costs were (non-significantly) higher compared with usual care. After excluding costs unlikely to be related to vestibular symptoms or internet-based VR, stand-alone VR became dominant over usual care in the cost-effectiveness analysis (ie, more effective and less costly). Internet-based VR is an easily accessible, effective form of treatment that could potentially improve care for a largely undertreated group of patients with a chronic vestibular syndrome in general practice. Based on the results of this economic evaluation, stand-alone VR should be preferred over blended VR.

**Contributors** ORM obtained the funding and coordinated the study. ORM, VAvV, JvdW, HvdH and JB were involved in the design of the study. VAvV collected the data. JB and APF analysed the data. VAvV and JB wrote the first substantial draft of the article and VAvV is the guarantor. ORM, JvdW, JB, APF and HvdH critically revised the manuscript. All authors read and approved the final manuscript.

**Funding** This study was funded by The Netherlands Organisation for Health Research and Development (ZonMw) (programme Quality in health care; grant number 839110015). The sponsors did not participate in the study design, data collection, analysis, interpretation, or the preparation or submission of this report.

**Competing interests** All authors have completed the Unified Competing Interest form (available on request from the corresponding author) and declare: support from The Netherlands Organisation for Health Research and Development (ZonMw); no financial relationships with any organisations that might have an interest in the submitted work in the previous three years, no other relationships or activities that could appear to have influenced the submitted work.

**Patient consent for publication** Not required.

**Ethics approval** The study protocol was approved by the Medical Ethics Committee of the VU University Medical Center. All participants included in the study provided written informed consent.

**Provenance and peer review** Not commissioned; externally peer reviewed.

**Data availability statement** Data are available upon reasonable request. De-identified individual participant data and data analysis plan available from the corresponding author on reasonable request.

**ORCID iD**
Vincent A van Vugt http://orcid.org/0000-0002-2746-4820

van Vugt VA, *et al. BMJ Open* 2020;**10**:e035583. doi:10.1136/bmjopen-2019-035583

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
