## [Reviewer comments · BMJ Open]

ARTICLE DETAILS

TITLE (PROVISIONAL)	The cost-effectiveness of internet-based vestibular rehabilitation with and without physiotherapy support for adults aged 50 and older with a chronic vestibular syndrome in general practice
AUTHORS	van Vugt, Vincent; Bosmans, J; Finch, Aureliano; van der Wouden, Johannes; van der Horst, Henriette; Maarsingh, Otto

VERSION 1 – REVIEW

REVIEWER	Dr. Ralf Strobl Institute for Medical Information Processing, Biometrics and Epidemiology - IBE German Center for Vertigo and Balance Disorders Ludwig-Maximilians-Universität München Germany
REVIEW RETURNED	14-Jan-2020

GENERAL COMMENTS	The authors evaluated the cost effectiveness of a internet-based vertigo rehabilitation tool (stand-alone and blended) to usual care. The study represent a secondary aim of the main study which evaluated the effectiveness of this program on vertigo. The study addresses a novel topic as vestibular rehabilitation is underrepresented in primary care and cost effectiveness is an important argument for better implementation. General: The outcome measure VSS-SF was already addressed in a recent publication [1] and does not need to be repeated here as a main outcome. Just use it, where needed for assessing cost-effectiveness. As the effect of the blended VT was shown to be worse than the stand-alone [1], it might be easier to concentrate on the results of the stand-alone arm and compare this to usual care. Abstract 1. Give numbers and p-value for difference.2. If not significant, do not use higher etc.3. Conclusion, if not founded by data do not report findings. Method 1. Use full words for first appearance of abbreviation ICVD2. Give more information on the calculation of the EQ5D.3. Describe, how QALYs can be interpreted.4. How was probability of cost-effectiveness calculated?5. What is time frame for “iMTA Medical and Productivity”? Does the questionnaire measure the absenteeism for the last 6 months?6. How were sociodemographic data measured?7. Which VSS-SF values do you report? Baseline, 3 months or 6 months. This is not clear from the text or the tables.
---

	Results 1. In general (abstract, result, discussion) be more cautious/conservative with the presentation of non-significant result, i.e. do not give hints for interpretation if no significance difference could be detected. Give the effect estimates in the description of the results. E.g. instead of writing: “For QALYs, both stand-alone and blended VR were more effective than usual care, but these differences were not statistically significant.” use “Stand-alone VR reported a mean QALY of 0.43, blended of 0.41 and usual care of 0.41. Differences of VR to usual care were not significant (0.02; CI [-0.0001; 0.04] and 0.01 [-0.01; 0.03]).” 2. If possible, give measure of uncertainty (confidence interval, standard deviations) 3. The results in Table 1 regarding VSS-SF do not fit the results of the ones in [1]. A reason might be the use of “pooled mean effects”. How are they calculated? Discussion 1. The intervention itself might have influenced the probability of the patients to stay away from work. This might explain why in the blended group absenteeism was twice as much as in usual care and 4 times as in stand-alone. [1] van Vugt, V. A., J. C. van der Wouden, R. Essery, L. Yardley, J. W. R. Twisk, H. E. van der Horst and O. R. Maarsingh (2019). "Internet based vestibular rehabilitation with and without physiotherapy support for adults aged 50 and older with a chronic vestibular syndrome in general practice: three armed randomised controlled trial." BMJ 367: l5922.
--	---

REVIEWER	Marinella D Galea The James J Peter VA Medical Center, Bronx NY
REVIEW RETURNED	23-Jan-2020

GENERAL COMMENTS	The article presents an interesting cost-analysis of different delivery methods for vestibular rehabilitation. The efforts to determine the costs specific to each modality are impressive as well as the discussion. The concepts of utilizing presentism in this context is very interesting. It is unfortunate that some of the results did not reach statistical significance. As mentioned by the authors, previous work has shown increased cost-effectiveness in the long term (12 months). The statistical analysis performed is beyond my expertise and I am unable to comment on that aspect of the manuscript.
---

VERSION 1 – AUTHOR RESPONSE

Reviewer #1:

The authors evaluated the cost effectiveness of an internet-based vertigo rehabilitation tool (stand-alone and blended) to usual care. The study represent a secondary aim of the main study which evaluated the effectiveness of this program on vertigo.

The study addresses a novel topic as vestibular rehabilitation is underrepresented in primary care and cost effectiveness is an important argument for better implementation.

General:

The outcome measure VSS-SF was already addressed in a recent publication [1] and does not need to be repeated here as a main outcome. Just use it, where needed for assessing cost-effectiveness. As the effect of the blended VT was shown to be worse than the stand-alone [1], it might be easier to concentrate on the results of the stand-alone arm and compare this to usual care.

Response: We agree the outcome measure VSS-SF should not be the main focus of this manuscript. For that reason, we present the calculated Quality-Adjusted Life-Years (QALYs) as the primary outcome measure. However, in order to optimally describe the cost-effectiveness of internet-based vestibular rehabilitation it is important to assess effectiveness both in terms of QALYs and improvement in VSS-SF, since the VSS-SF is one of the most frequently used patient-reported outcome measures in vestibular medicine.

We believe it is important to assess both the cost-effectiveness of stand-alone VR and blended VR. Stand-alone VR and blended VR were both shown to be more effective than usual care in reducing vestibular symptoms. A qualitative interview study we conducted with blended VR participants and physiotherapists indicated that the physiotherapy visits provided personal attention, helped patients more safely execute exercises, and improved patients' adherence to therapy [Van Vugt, Personal communication, 2020]. Although these additional benefits did not result in a higher effect size for blended VR, some patients may be more suited to receive blended VR than stand-alone VR. We also performed a prediction study among the present study population that showed that it is difficult to predict treatment success of internet-based VR and it remains unclear who should be treated with stand-alone VR or blended VR [Van Vugt, Personal communication, 2020]. Therefore, the decision to offer stand-alone or blended VR should be based on availability, patient preference and cost-effectiveness. Consequently, we have described the cost-effectiveness of both stand-alone and blended VR in this manuscript.

Changes:

None

Abstract

1. Give numbers and p-value for difference.

Response: We have added the numbers to the abstract. If you will permit us, we would like to report the 95% confidence interval instead of the p-values. We prefer 95% confidence intervals over p-values, because they provide more information about the precision of the results.

Changes:

Page 2

Abstract

Mean societal costs in both the stand-alone and blended VR groups were statistically non-significantly higher than in the UC group (mean difference (MD) €504, 95%-CI -1082 to 2268; and €916, 95%-CI -663 to 2596). Both stand-alone and blended VR groups reported non-significantly more QALYs than the UC group (MD 0.02, 95%-CI -0.00 to 0.04; and 0.01, 95%-CI -0.01 to 0.03), and significantly better VSS-SF scores (MD 3.8 points, 95%-CI 1.7 to 6.0; and 3.3 points, 95%-CI 1.3 to 5.2).

2. If not significant, do not use higher etc.

Response: We have added numbers and 95% confidence intervals to the abstract to present the results more clearly. We do not agree that a lack of statistical significance prevents us from using a term like “higher”. The RCT was not powered to detect differences in cost-effectiveness, but to investigate a clinically relevant difference in VSS-SF improvement. As is often the case in economic evaluations conducted alongside RCTs, the sample size was insufficient to determine statistically significant differences in costs. Describing the direction of the effects provides more information than only stating that no significant differences were found. We do agree that the results should therefore be interpreted with care.

3. Conclusion, if not founded by data do not report findings.

Response: We agree with this statement. However, as we explained in response to the previous question, we believe that our current conclusion is founded by data.

Method

1. Use full words for first appearance of abbreviation ICVD

Response: Thank you, we adjusted this in the revised manuscript.

Changes:

Page 6

Participants

In short, we recruited participants from 59 general practices in The Netherlands. Patients with chronic vestibular syndrome according to the International Classification of Vestibular Disorders (ICVD) were eligible for the study.

2. Give more information on the calculation of the EQ5D.

Response: We provided more information on the calculation of the EQ5D outcome measure.

Changes:

Page 8

Measures

EQ-5D-5L health states were converted to utility scores using the Dutch EQ-5D-5L tariff(29). For all five dimensions, a subtraction is done from the utility score if the participant experiences any problems on that domain. We calculated Quality-Adjusted Life-Years (QALYs) by multiplying the utility of a specific health state with the time spent in that health state.

3. Describe, how QALYs can be interpreted.

Response: We described the way in which QALYs can be interpreted in more detail.

Changes:

Page 8

Measures

One QALY is equivalent to one life-year in perfect health. A gain in QALYs can therefore be interpreted as an improvement in the quantity and/or quality of life.

4. How was probability of cost-effectiveness calculated?

Response: The probability of cost-effectiveness was calculated by determining the proportion of bootstrapped cost-effect pairs below the willingness-to-pay threshold for each possible willingness-to-pay threshold.

Changes:

Page 10

Cost-effectiveness analyses

The probability of cost-effectiveness was calculated by determining the proportion of bootstrapped cost-effect pairs below the willingness-to-pay threshold for each possible willingness-to-pay threshold.

5. What is time frame for “iMTA Medical and Productivity”? Does the questionnaire measure the absenteeism for the last 6 months?

Response: The iMCQ and iPCQ were assessed at 3 and 6 months and had a recall period of 3 months. The total time frame in which we measured absenteeism is therefore 6 months. We clarified this in the manuscript.

Changes:

Page 8

Measures

We measured costs from a societal perspective, using the iMTA Medical and Productivity Cost Questionnaires at 3 and 6 months of follow-up.(34, 35) Both questionnaires had a recall period of 3 months.

6. How were sociodemographic data measured?

Response: Participants filled out a questionnaire about their age, sex, level of education and living situation at baseline. The sociodemographic data are described in more detail in a previous publication (Van Vugt, BMJ, 2019).

Changes:

None

7. Which VSS-SF values do you report? Baseline, 3 months or 6 months. This is not clear from the text or the tables.

Response: We report the VSS-SF value at six months, the primary outcome measure of our RCT (Van Vugt, BMJ, 2019). We have added information to the methods and results sections to describe this more clearly.

Changes:

Page 8

Measures

In this cost-effectiveness study, we used the VSS-SF scores at 6 months follow-up and the number of participants with a clinically relevant improvement during the 6-month trial period as measures of effectiveness.

Page 12

Effects

Both stand-alone and blended VR participants reported statistically significantly lower VSS-SF score at 6 months follow-up compared to usual care (mean difference 3.8 points, 95% CI 1.7 to 6.0; and 3.3 points, 95% CI 1.3 to 5.2).

Results

1. In general (abstract, result, discussion) be more cautious/conservative with the presentation of non-significant result, i.e. do not give hints for interpretation if no significance difference could be detected. Give the effect estimates in the description of the results. E.g. instead of writing: "For QALYs, both stand-alone and blended VR were more effective than usual care, but these differences were not statistically significant." use "Stand-alone VR reported a mean QALY of 0.43, blended of 0.41 and usual care of 0.41. Differences of VR to usual care were not significant (0.02; CI [-0.0001; 0.04] and 0.01 [-0.01; 0.03])."

Response: Thank you. We have adjusted the results section and now report more effect estimates in the text of the manuscript.

Changes:

See revised results section.

2. If possible, give measure of uncertainty (confidence interval, standard deviations

Response: We have added these measures of uncertainty throughout the results section.

Changes:

See response to previous question.

3. The results in Table 1 regarding VSS-SF do not fit the results of the ones in [1]. A reason might be the use of "pooled mean effects". How are they calculated?

Response: Yes, we used different statistical methods for the clinical and cost-effectiveness analyses. In the clinical effectiveness analysis we used a linear mixed models analysis. This technique can account for repeated measures within an individual and is also capable of handling missing data in a longitudinal dataset without the need to perform multiple imputations. However, to conduct the cost-effectiveness analysis performing multiple imputation was necessary. The effectiveness analysis for this paper was therefore also conducted in an imputed dataset which explains the slight difference in results reported. We added a note to Table 1 to explain this difference.

Changes:

Page 23

Table 1

* Due to a different method of analysis VSS-SF scores slightly differ from the previously reported clinical effectiveness analysis. (25)

Discussion

1. The intervention itself might have influenced the probability of the patients to stay away from work. This might explain why in the blended group absenteeism was twice as much as in usual care and 4 times as in stand-alone.

Response: Since absenteeism was lower for participants in the stand-alone VR group compared to usual care (€241 versus €481) the effects do not seem to be the same for all forms of internet-based VR. We agree that physiotherapy visits in blended VR may have influenced the probability of

participants to stay away from work. Because we conducted our qualitative interview study with blended VR participants and physiotherapists before we performed these cost-effectiveness analyses, we were not able to investigate this hypothesis yet [Van Vugt, Personal communication, 2020].

Changes:
None

Reviewer #2

The article presents an interesting cost-analysis of different delivery methods for vestibular rehabilitation. The efforts to determine the costs specific to each modality are impressive as well as the discussion. The concepts of utilizing presentism in this context is very interesting. It is unfortunate that some of the results did not reach statistical significance. As mentioned by the authors, previous work has shown increased cost-effectiveness in the long term (12 months). The statistical analysis performed is beyond my expertise and I am unable to comment on that aspect of the manuscript.

Response: Thank you for your compliments about the paper.

VERSION 2 – REVIEW

REVIEWER	Ralf Strobl Institute for Medical Information Processing, Biometrics and Epidemiology - IBE German Center for Vertigo and Balance Disorders Ludwig-Maximilians-Universität München Germany
REVIEW RETURNED	13-Aug-2020
GENERAL COMMENTS	I would like to thank the authors for their thoughtful revision of the manuscript and for giving their reasoning behind all changes. Especially, giving confidence intervals is preferable over simple p-values! This gives the reader the information at hand to interpret the results with adequate care.